# SALMONN: Towards Generic Hearing Abilities for Large Language Models

**Changli Tang**[1][*] **Wenyi Yu**[1][*] **Guangzhi Sun**[1], **Xianzhao Chen**[2], **Tian Tan**[2]
**Wei Li**[2], **Lu Lu**[2], **Zejun Ma**[2], **Chao Zhang**[1][†]
Department of Electronic Engineering, Tsinghua University[1]
ByteDance[2]
{tcl20,ywy22}@mails.tsinghua.edu.cn, cz277@tsinghua.edu.cn

## Abstract

Hearing is arguably an essential ability of artificial intelligence (AI) agents in the physical world, which refers to the perception and understanding of general auditory information consisting of at least three types of sounds: speech, audio events, and music. In this paper, we propose SALMONN, a speech audio language music open neural network, built by integrating a pre-trained text-based large language model (LLM) with speech and audio encoders into a single multimodal model. SALMONN enables the LLM to directly process and understand general audio inputs and achieve competitive performances on a number of speech and audio tasks used in training, such as automatic speech recognition and translation, auditory-information-based question answering, emotion recognition, speaker verification, and music and audio captioning *etc.* SALMONN also has a diverse set of emergent abilities unseen in the training, which includes but is not limited to speech translation to untrained languages, speech-based slot filling, spoken-query-based question answering, audio-based storytelling, and speech audio co-reasoning *etc.* The presence of cross-modal emergent abilities is studied, and a novel few-shot activation tuning approach is proposed to activate such abilities. To our knowledge, SALMONN is the first model of its type and can be regarded as a step towards AI with generic hearing abilities. The source code, model checkpoints and data are available at https://github.com/bytedance/SALMONN.

## 1 Introduction

Text-based large language models (LLMs) (Brown et al., 2020; Touvron et al., 2023; Chiang et al., 2023; Anil et al., 2023; Du et al., 2022) have demonstrated remarkable and even human-level performance in many natural language processing (NLP) tasks (OpenAI, 2023). Meanwhile, instruction tuning (Wei et al., 2022a; Chung et al., 2022; Ouyang et al., 2022; Peng et al., 2023), where data is organised as pairs of user instruction (or prompt) and reference response, has emerged as an LLM training paradigm that allows LLMs to follow open-ended user instructions. There is a burgeoning research interest in empowering LLMs with multimodal perception abilities. Recent studies focus on connecting LLMs with either the encoder of one additional type of input, such as image (Li et al., 2023a; Alayrac et al., 2022; Dai et al., 2023), silent video (Maaz et al., 2023; Chen et al., 2023b; Zhao et al., 2022), audio events (Gong et al., 2023b; Lyu et al., 2023) or speech (Chen et al., 2023a), or the encoders of multiple input types together (Su et al., 2023; Zhang et al., 2023b). A connection module and LLM adaptors can be used to align the encoder output spaces with the LLM input space, which are often trained by cross-modal pre-training and instruction tuning.

In this paper, we propose a speech audio language music open neural network (SALMONN), which is a single audio-text multimodal LLM that can perceive and understand three basic types of sounds including speech, audio events, and music. To enhance the performance on both speech and non-speech audio tasks, SALMONN uses a dual encoder structure consisting of a speech encoder from the Whisper speech model (Radford et al., 2023) and a BEATs audio encoder (Chen et al., 2023c). A

---

[*]Equal contribution
[†]Corresponding author

window-level query Transformer (Q-Former) (Li et al., 2023a) is used as the connection module to convert a variable-length encoder output sequence to a variable number of augmented audio tokens input to the Vicuna LLM (Chiang et al., 2023) and can achieve audio-text alignment with high temporal resolution. The low-rank adaptation (LoRA) approach (Hu et al., 2022) is applied to Vicuna as a cross-modal adaptor to align Vicuna's augmented input space with its output space and further improve its performance. A number of speech, audio, and music tasks are used in the cross-modal pre-training and instruction tuning stages of the window-level Q-Former and LoRA. The resulting multimodal LLMs can be confined to the specific types of tasks used in instruction tuning, particularly speech recognition and audio captioning, and exhibit limited or no *cross-modal emergent abilities*, which we refer to as the *task over-fitting* issue. In this paper, cross-modal emergent abilities refer to the abilities to perform cross-modal tasks unseen in training, which are essentially the emergent abilities of LLMs (Wei et al., 2022b) that are lost during instruction tuning. As a solution, we propose an extra few-shot activation tuning stage so that SALMONN regains the emergent abilities of LLMs and alleviates the considerable *catastrophic forgetting* to the trained tasks.

In order to evaluate SALMONN's cognitive hearing abilities, a wide range of speech, audio events, and music benchmarks are used. The tasks can be divided into three levels. The first level benchmarks eight tasks that are trained in instruction tuning, such as speech recognition, translation and audio captioning, while the other two levels benchmark untrained tasks. The second level includes five speech-based NLP tasks, such as translation to untrained languages and slot filling, which relies on multilingual and high-quality alignments between speech and text tokens. The last level of tasks, including audio-based storytelling and speech audio co-reasoning *etc*, requires understanding not only speech but also non-speech auditory information. Experimental results show that SALMONN as a single model can perform all these tasks and achieve competitive performance on standard benchmarks, which reveals the feasibility of building artificial intelligence (AI) that can "hear" and understand *general audio inputs* consisting of mixtures of speech, audio events, and music.

The main contribution of this paper can be summarised as follows.

- We propose SALMONN, the first multimodal LLM that can perceive and understand general audio inputs with speech, audio events, and music, to the best of our knowledge.

- We study the presence of cross-modal emergent abilities by playing with the LoRA scaling factor, and propose a cheap activation tuning method as an extra training stage that can activate the cross-modal emergent abilities and alleviate catastrophic forgetting to tasks seen in training.

- We evaluate SALMONN on a range of tasks reflecting a degree of generic hearing abilities, and propose two novel tasks, audio-based storytelling and speech audio co-reasoning.

## 2 RELATED WORK

LLMs, as text-based dialogue models, have a fundamental connection with speech, and several studies have attempted to extend LLMs to support direct speech inputs with a connection module (Chen et al., 2023a; Wu et al., 2023; Fathullah et al., 2023; Yu et al., 2023; Huang et al., 2023a). To avoid the LLMs having overly long input speech token sequences caused by long-form speech inputs, different frame-rate reduction approaches have been developed, including stacking-based fixed-rate reduction approach (Fathullah et al., 2023; Yu et al., 2023), speech-recognition-based variable frame-rate reduction approach (Wu et al., 2023; Chen et al., 2023a), and Q-Former-based approach with a fixed number of output frames (Yu et al., 2023) *etc*. When LLM-based speech synthesis is also considered, the LLM output space can be augmented with speech tokens as well, such as SpeechGPT (Zhang et al., 2023a) and AudioPaLM (Rubenstein et al., 2023).

Unlike speech, audio event inputs are often treated as fixed-sized spectrogram images that can be processed using visual-language LLM methods without explicitly modelling temporal correlations (Gong et al., 2023a;b; Zhang et al., 2023b). These methods are therefore unable to handle speech. Although Lyu et al. (2023) uses the speech encoder from the Whisper model, only audio event inputs are supported, which indicates the difficulty of the joint modelling of speech and audio events. Without using LLMs, Narisetty et al. (2022) studies achieving speech recognition and audio captioning separately using the same model. Regarding music inputs, Liu et al. (2023) integrates the MERT music encoder (Li et al., 2023b) with an LLM for music understanding tasks. AudioGPT allows a text-based LLM to process speech, audio events, and music by interacting with other models in

a pipeline based on a set of pre-defined tasks (Huang et al., 2023b). Compared with AudioGPT, SALMONN is an end-to-end model with cross-modal emergent abilities for open-ended tasks.

In addition to audio, multimodal LLMs are more widely studied in visual modalities, such as image (Zhu et al., 2023; Li et al., 2023a), video (Maaz et al., 2023; Chen et al., 2023b) and audio-visual (Su et al., 2023; Lyu et al., 2023; Sun et al., 2023). Modality alignment in those models is often achieved via either a fully connected layer or an attention-based module. In particular, the Q-Former structure (Li et al., 2023a) used by SALMONN is commonly applied to visual modalities, such as in MiniGPT-4 (Zhu et al., 2023), InstructBLIP (Dai et al., 2023), Video-LLaMA (Zhang et al., 2023b).

## 3 METHODOLOGY

The model architecture of SALMONN is introduced in Section 3.1. Our training method is presented in Section 3.2, which includes the pre-training and fine-tuning stages, and the proposed activation tuning stage as a solution to the task over-fitting issue.

### 3.1 MODEL ARCHITECTURE

The model architecture of SALMONN is shown in Fig. 1. The output features of the two complementary auditory encoders are synchronised and combined. Q-Former is used as the connection module and applied at the frame level, whose output sequence is integrated with the text instruction prompt and fed into the LLM with LoRA adaptors to generate the text response.

**Dual Auditory Encoders:** A speech encoder from OpenAI's Whisper model (Radford et al., 2023) and a non-speech BEATs audio encoder (Chen et al., 2023c) are used. The Whisper model is trained for speech recognition and translation based on a large amount of weakly supervised data, whose encoder output features are suitable to model speech and include information about the background noises (Gong et al., 2023a). BEATs is trained to extract high-level non-speech audio semantics information using iterative self-supervised learning. The input audio is first tokenised then masked and predicted in training. The tokeniser is updated by distilling the semantic knowledge of the audio tokens (Chen et al., 2023c). Therefore, the resulting auditory features of these two encoders are complementary and suitable for general audio inputs with both speech and non-speech information.

Since both encoders have the same output frame rate of 50Hz, the concatenated output features are

$$\mathbf{Z} = \text{Concat}(\text{Encoder}_{\text{whisper}}(\mathbf{X}), \text{Encoder}_{\text{beats}}(\mathbf{X})), \tag{1}$$

where $\mathbf{X}$ is a variable-length general audio input sequence, $\text{Encoder}_{\text{whisper}}(\cdot)$ and $\text{Encoder}_{\text{beats}}(\cdot)$ are the Whisper and BEATs encoder, $\text{Concat}(\cdot)$ is the frame-by-frame concatenation operation along the feature dimension, $\mathbf{Z}$ is the concatenated encoder output sequence with $T$ frames.

**Window-level Q-Former:** The Q-Former structure is commonly used to convert the output of an image encoder into a fixed number of textual input tokens of an LLM (Li et al., 2023a), which requires modification when applied to handle audio inputs of variable lengths. Specifically, regarding the encoder output of an input image $l$ as a $\mathbf{Z}_l$, Q-Former employs a fixed number of $N$ trainable queries $\mathbf{Q}$ to transform $\mathbf{Z}_l$ into $N$ textual tokens $\mathbf{H}_l$ using a number of stacked Q-Former blocks. A Q-Former block is similar to a Transformer decoder block (Vaswani et al., 2017), apart from the use of a fixed number of trainable static queries $\mathbf{Q}$ in the first block and the removal of the causal masks from the self-attention layers. In this way, Q-Former allows the queries in $\mathbf{Q}$ to refer to each other first using a self-attention layer and then interact with $\mathbf{Z}_l$ using cross-attention layers.

Regarding a variable-length general audio input with $\mathbf{Z} = [\mathbf{Z}_t]_{t=1}^{T}$, by segmenting $\mathbf{Z}$ into $L$-sized windows and padding the last window with zeros, it becomes $[\{\mathbf{Z}_t\}_{t=(l-1)\times L+1}^{l\times L}]_{l=1}^{\lceil T/L \rceil}$, instead of using Q-Former at the sequence level to convert the entire $\mathbf{Z}$ into $N$ textual tokens, SALMONN uses Q-Former at the window level as if the encoder output frames stacked in each window were an image. As a result, the textual token sequence $\mathbf{H}$ becomes

$$[\mathbf{H}_l]_{l=1}^{\lceil T/L \rceil} = [\text{Q-Former}(\mathbf{Q}, \mathbf{Z}_l)]_{l=1}^{\lceil T/L \rceil}, \tag{2}$$

where $\text{Q-Former}(\cdot)$ is the Q-Former function and $\mathbf{H}$ has $\lceil T/L \rceil \times N$ textual tokens. The window-level Q-Former uses a variable number of textual tokens and is more efficient for variable-length

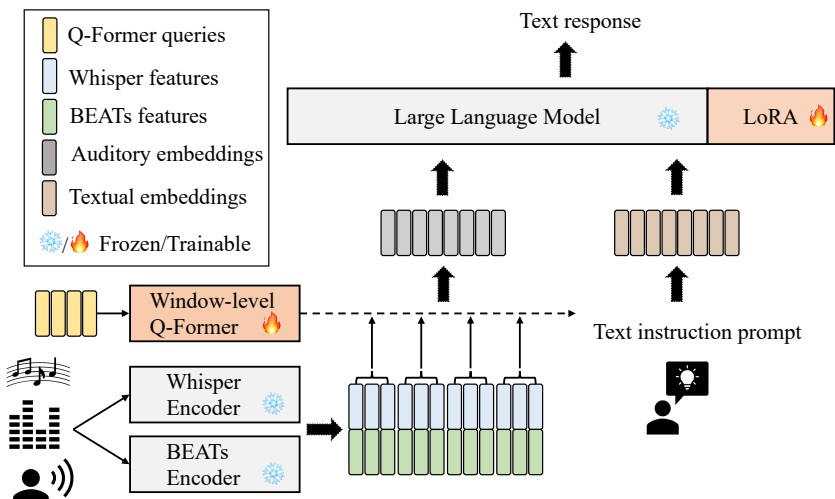

Figure 1: The model architecture of SALMONN. A window-level Q-Former is used as the connection module to fuse the outputs from a Whisper speech encoder and a BEATs audio encoder as augmented audio tokens, which are aligned with the LLM input space. The LoRA adaptor aligns the augmented LLM input space with its output space. The text prompt is used to instruct SALMONN to answer open-ended questions about the general audio inputs and the answers are in the LLM text responses. The LLM and encoders are kept frozen while the rest can be updated in training.

sequences. In addition, $\mathbf{H}$ is enforced to have a monotonic alignment with $\mathbf{Z}$, resulting in better temporal resolution which is important for speech recognition.

**LLM and LoRA:** A pre-trained Vicuna LLM is used in this work (Chiang et al., 2023) which is a LLaMA LLM (Touvron et al., 2023) fine-tuned to follow instructions. LoRA (Hu et al., 2022) is a widely used parameter-efficient fine-tuning method for LLM adaptation, which is used in SALMONN to adapt the query and value weight matrices in the self-attention layers of Vicuna. In this work, LoRA is trainable while Vicuna is not.

## 3.2 TRAINING METHOD

A three-stage cross-modal training method of SALMONN is introduced in this section. Besides the pre-training and instruction tuning stages used by recent visual LLMs (Dai et al., 2023; Zhang et al., 2023b), an additional activation tuning stage is proposed to resolve the issue of over-fitting to the speech recognition and audio captioning tasks in instruction tuning.

**Pre-training Stage:** To mitigate the gap between the pre-trained parameters (LLM and encoders) and the randomly initialised parameters (connection module and adaptor), a large amount of speech recognition and audio captioning data is used to pre-train the window-level Q-Former and LoRA. These two tasks contain key auditory information about the contents of speech and non-speech audio events, and both do not require complex reasoning and understanding and therefore can help SALMONN to learn high-quality alignment between the auditory and textual information.

**Instruction Tuning Stage:** Similar to NLP (Wei et al., 2022a) and visual-language (Dai et al., 2023), audio-text instruction tuning with a list of supervised speech, audio event, and music tasks, as shown in Section 4.2 and Table 1, is used as the second stage of SALMONN training. The tasks are selected based on their importance (*e.g.* speech recognition and audio captioning) and the indispensability of having such ability in tests (*e.g.* overlapping speech recognition, phone recognition and music captioning). The instruction prompts are generated based on the texts paired with the audio data.

**Task Over-fitting:** Although SALMONN built with only the first two stages of training can produce competitive results on the tasks trained in instruction tuning, it exhibits limited or almost no ability to perform untrained cross-modal tasks, especially the tasks that require cross-modal co-reasoning abilities. In particular, the model sometimes violates the instruction prompts and generates irrelevant responses as if it received an instruction related to a task commonly seen in training (*e.g.* speech

recognition). We refer to this phenomenon as *task over-fitting*. A theoretical analysis of the issue is presented here and detailed experimental verification is provided in Sections 5.3 and 5.4.

We attribute task over-fitting to two reasons. First, compared to the text-only data used in LLM training, only simpler instruction prompts are used in our cross-modal instruction tuning (Wei et al., 2022a) and the resulting responses are not as complex and diverse. Meanwhile, some tasks included in instruction tuning, in particular speech recognition and audio captioning, have more deterministic outputs than the other tasks, such as speech and audio question answering. These two reasons combined to cause the intrinsic conditional language model (LM) to bias to an ill-formed distribution with poor generalisation ability, which prohibits SALMONN from performing untrained cross-modal tasks. More specifically, at test time the response text sequence $\hat{\mathbf{Y}}$ of a test input $\mathbf{X}$ given a new instruction prompt $\mathbf{I}$ can be generated according to $\hat{\mathbf{Y}} = \arg\max_{\mathbf{Y}} P_{\mathbf{\Lambda}}(\mathbf{Y}|\mathbf{X}, \mathbf{I})$, which is also the objective to maximise in training. Using the *Bayes' Rule*, there is

$$P_{\mathbf{\Lambda}}(\mathbf{Y}|\mathbf{X}, \mathbf{I}) = P_{\mathbf{\Lambda}}(\mathbf{Y}|\mathbf{X}) \cdot P_{\mathbf{\Lambda}}(\mathbf{I}|\mathbf{Y}, \mathbf{X})/P_{\mathbf{\Lambda}}(\mathbf{I}|\mathbf{X}). \tag{3}$$

Since only limited text responses are seen in SALMONN training, the intrinsic conditional LM $P_{\mathbf{\Lambda}}(\mathbf{Y}|\mathbf{X})$ is biased towards the $\mathbf{Y}$ sequences that are strongly-aligned with $\mathbf{X}$, such as the transcriptions of automatic speech recognition (ASR) and automatic audio captioning (AAC) tasks, especially simple and short transcriptions. From Eqn. (3), this causes $\mathbf{I}'$, zero-shot instructions with more diverse responses, to have small $P_{\mathbf{\Lambda}}(\mathbf{Y}|\mathbf{X}, \mathbf{I}')$.

**Activation Tuning Stage:** An effective approach to alleviating task over-fitting is to regularise the intrinsic conditional LM $P_{\mathbf{\Lambda}}(\mathbf{Y}|\mathbf{X})$. An easy way to achieve this is to fine-tune SALMONN on tasks with longer and more diverse responses, such as auditory-information-based question answering and storytelling. Paired training data for such tasks can be generated based on the text paired with speech recognition or audio and music caption data, either manually by human annotators or automatically by prompting a text-based LLM.

We use an efficient approach that can enable SALMONN to generate long and diverse responses for zero-shot instructions by simply reducing the scaling factor of the LoRA adaptor. This is an alternative way to regularise $P_{\mathbf{\Lambda}}(\mathbf{Y}|\mathbf{X})$ since the intrinsic conditional LM can only be learned with the window-level Q-Former and LoRA, since they are the only modules that are updated in training. The effect of reducing the LoRA scaling factor can be found in Section 5.2, which can indeed activate question-answering and storytelling abilities and produce long and diversified responses but also considerably degrade the results on the trained tasks. To retain competitive results while restoring the cross-modal emergent abilities, we propose to use the responses generated by SALMONN with a discounted LoRA scaling factor to perform the third stage of fine-tuning termed activation tuning. Experimental results showed later in Section 5.4 demonstrate that activation tuning is an efficient and effective few-shot self-supervised training approach.

## 4 EXPERIMENTAL SETUP

### 4.1 MODEL SPECIFICATIONS

SALMONN uses the encoder part of Whisper-Large-v2 (Radford et al., 2023) model as the speech encoder, the fine-tuned BEATs (Chen et al., 2023c) encoder as the audio encoder, and a Vicuna LLM with 13 billion parameters (Chiang et al., 2023) as the backbone LLM. For the window-level Q-Former, we use $N = 1$ resulting in only one trainable query, and use $L = 17$ which is approximately 0.33 seconds per window. This leads to 88 textual tokens output by Q-Former for a 30-second audio. Regarding the hyper-parameters of LoRA (Hu et al., 2022), we set the rank to 8 and the scaling factor to 4.0. Only the parameters of Q-Former and LoRA are updated in training, which resulted in $\sim 33$ million (M) parameters that is $\sim 0.24\%$ of the entire SALMONN model.

### 4.2 DATA SPECIFICATIONS

The three-stage training proposed in Section 3.2 is used. The data used for the first pre-training stage consists of both 960-hour LibriSpeech training set (Panayotov et al., 2015) and 1000-hour GigaSpeech M-set (Chen et al., 2021) for speech recognition, as well as 2800-hour WavCaps (Mei et al., 2023) (with audio clips longer than 180 seconds removed), AudioCaps (Kim et al., 2019) and Clotho (Drossos et al., 2020) datasets for audio captioning.

The second instruction tuning stage involves multiple tasks, including ASR, automatic speech translation (AST), AAC, phone recognition (PR), emotion recognition (ER), music captioning (MC), overlapped speech recognition (OSR), speaker verification (SV), gender recognition (GR) and speech question answering (SQA), audio question answering (AQA) and music question answering (MQA). In the SQA, AQA and MQA tasks, the questions are generated based on the text caption labels using ChatGPT, and the model needs to provide answers based on the general audio input and the text prompt with a question. The data used in this stage is listed in Table 1, where "En2Zh" refers to AST from English to Chinese.

For the final activation tuning stage, twelve stories were written based on the audio clips by SALMONN with a reduced LoRA. Then the model is trained using teacher-forcing-based cross-entropy training for 12 steps, with each step using only one story sample, when activating the cross-modal emergent abilities of SALMONN.

Table 1: Training data used in the cross-modal instruction tuning stage.

| Task | Data Source | #Hours | #Samples |
|------|-------------|--------|----------|
| ASR | LibriSpeech + GigaSpeech | 960 + 220 | 280K + 200K |
| En2Zh | CoVoST2-En2Zh (Wang et al., 2021) | 430 | 290K |
| AAC | AudioCaps + Clotho | 130 + 24 | 48K + 4K |
| PR | LibriSpeech | 960 | 280K |
| ER | IEMOCAP Session 1-4 (Busso et al., 2008) | 5 | 4K |
| MC | MusicCaps (Agostinelli et al., 2023) | 14 | 3K |
| OSR | LibriMix (Cosentino et al., 2020) | 260 | 64K |
| SV | VoxCeleb1 (Nagrani et al., 2019) | 1200 | 520K |
| GR | LibriSpeech | 100 | 28K |
| SQA | LibriSpeech | 960 | 280K |
| AQA | WavCaps + AudioCaps | 760 + 130 | 270K + 48K |
| MQA | MillionSong[1] + MusicNet (Thickstun et al., 2017) | 400 + 3 | 48K + 0.3K |
| **Total** | – | ∼4400 | ∼2.3M |

## 4.3 TASK SPECIFICATIONS

Since text LLMs have the abilities of zero-shot learning via instruction tuning (Wei et al., 2022a), the emergence of such abilities is expected when high-quality cross-modal alignment is achieved when connecting the backbone text-based LLM with multimodal encoders. To evaluate the zero-shot cross-modal emergent abilities of SALMONN, 15 types of speech, audio, and music tasks are selected and divided into three different levels.

**Task level 1** consists of the tasks used in instruction tuning and are therefore easiest for SALMONN to perform. The list of such tasks and their training data are given in Section 4.2, and the evaluation metrics for each task are presented in Table 2.

**Task level 2** includes untrained tasks and is therefore more difficult than level 1. The level 2 tasks are speech-based NLP tasks including speech keyword extracting (KE), which evaluates the accuracy of the keywords extracted based on the speech content; spoken-query-based question answering (SQQA), which evaluates the common sense knowledge retrieved based on the question in speech; speech-based slot filling (SF) that evaluates the accuracy of the required slot values, usually named entities, obtained from the speech content. Two AST tasks, En2De (English to German) and En2Ja (English to Japanese) are also included, which are also considered as cross-modal emergent abilities since only En2Zh is trained in instruction tuning. Vicuna, the backbone LLM of SALMONN, can perform all level 2 tasks based on speech transcriptions. Therefore, SALMONN is to achieve such tasks based on speech in a fully end-to-end way without requiring any explicit speech recognition.

**Task level 3** has the most difficult tasks including audio-based storytelling (Story) and speech audio co-reasoning (SAC). Audio-based storytelling is to write a meaningful story based on the auditory information from the general audio inputs. SAC requires the model to understand a spoken question embedded in the input audio clip, find evidence from the background audio events or music, and

---

[1]https://www.kaggle.com/datasets/undefinenull/million-song-dataset-spotify-lastfm

reason from it to answer the question. Both level 3 tasks are new tasks that are first proposed in this paper, to the best of our knowledge, which requires SALMONN to perceive speech, audio, and music, and to understand and reason based on the auditory information in a fully end-to-end way.

Table 2 lists all test sets and their evaluation metrics. *Following rate* (FR) is an extra metric used for some level 2 and level 3 tasks, which measures the percentage that SALMONN can successfully follow the instructions. FR is considered since the selected tasks are complex and easier to suffer from the violation of instructions caused by task over-fitting. It is worth noting that we only calculate the diversity metric of the Story task by counting the number of different words in the story, which simply represents the richness of the story instead of the quality.

Table 2: Test sets, metrics, and sources of the reference values used in the three levels of tasks. The speech data used in SQQA and KE are synthesised using a commercial text-to-speech product. For reference values: "Whisper" refers to using Whisper-Large-v2 for ASR, "Whisper + Vicuna" refers to feeding the Whisper-Large-v2 ASR output transcriptions into Vicuna, and the rest are the state-of-the-art results to the best of our knowledge.

| Task | Test Data | Eval Metrics | Reference Value |
|---|---|---|---|
| ASR | LibriSpeech test-clean/-other, | %WER | Whisper |
| ASR | GigaSpeech test | %WER | Whisper |
| En2Zh | CoVoST2-En2Zh | BLEU4 | (Wang et al., 2021) |
| AAC | AudioCaps | METEOR \| SPIDEr | (Mei et al., 2023) |
| PR | LibriSpeech test-clean | %PER | WavLM (Chen et al., 2022) |
| ER | IEMOCAP Session 5 | Accuracy | (Wu et al., 2021) |
| MC | MusicCaps | BLEU4, RougeL | (Doh et al., 2023) |
| OSR | LibriMix | %WER | (Huang et al., 2023c) |
| SV | Voxceleb1 | Accuracy | - |
| En2De | CoVoST2-En2De | BLEU4 | Whisper + Vicuna |
| En2Ja | CoVoST2-En2Ja | BLEU4 | Whisper + Vicuna |
| KE | Inspec (Hulth, 2003) | Accuracy | Whisper + Vicuna |
| SQQA | WikiQA (Yang et al., 2015) | Accuracy (FR) | Whisper + Vicuna |
| SF | SLURP (Bastianelli et al., 2020) | Accuracy (FR) | Whisper + Vicuna |
| Story | AudioCaps | Diversity (FR) | – |
| SAC | In-house Data | Accuracy (FR) | – |

## 5 EXPERIMENTAL RESULTS

### 5.1 FULL RESULTS ON ALL 15 TASKS

The results on all 15 tasks produced by SALMONN are shown in Table 3. From the results, SALMONN, without or with activation tuning, can produce competitive results on all level 1 tasks. However, the model without activation tuning suffers severely from task over-fitting and can barely perform level 2 and level 3 tasks. In particular, in SQQA, Story, and SAC, where multimodal interactions are emphasised, SALMONN without activation tuning can hardly follow the instructions. The FRs of performing SQQA, SF, Story and SAC tasks improve considerably with activation tuning.

Fig. 2 illustrates the trends of model performance change on ASR & PR, SQQA, Story, and SAC at different training steps of activation tuning, which reveals that activation tuning only needs a few training samples and steps to activate the emergent abilities: the results of ASR and PR remain almost unchanged, while the results of SQQA, Story and SAC have an emergent trend.

More detailed analysis about the results are discussed in Appendix B due to page limit.

### 5.2 DISCOUNTING LORA SCALING FACTOR

This section explores the influence of the use of test-time discounting of the LoRA scaling factor for alleviating the task over-fitting issue without activation tuning. As shown in Fig. 3, when the LoRA scaling factor decreases to around 2.0, *i.e.* to half of its original value, the model suddenly emerges

Table 3: Results of all 15 tasks produced by SALMONN without & with activation tuning (w/o & w/ Activation). The ASR results are presented in a tuple with three the %WERs evaluated on 3 test sets, namely (LibriSpeech test-clean, LibriSpeech test-other, GigaSpeech).

| Method | ASR↓ | En2Zh↑ | AAC↑ | PR↓ | ER↑ | MC↑ | OSR↓ | SV↑ |
|---|---|---|---|---|---|---|---|---|
| w/o Activation | (2.1, 4.9, 9.1) | 34.4 | 25.6 \| 47.6 | 4.2 | 0.63 | 3.5, 22.1 | 20.7 | 0.93 |
| w/ Activation | (2.1, 4.9, 10.0) | 33.1 | 24.0 \| 40.3 | 4.2 | 0.69 | 5.5, 21.8 | 23.0 | 0.94 |
| Reference Value | (2.2, 5.1, 9.2) | 38.9 | 25.0 \| 48.5 | 3.1 | 0.81 | 6.1, 21.5 | 7.6 | - |

(a) Results of the level 1 tasks.

| Method | En2De↑ | En2Ja↑ | KE↑ | SQQA↑ | SF↑ | Story↑ | SAC↑ |
|---|---|---|---|---|---|---|---|
| w/o Activation | 19.7 | 22.0 | 0.30 | 0.19 (0.29) | 0.33 (0.77) | 7.77 (0.00) | 0.02 (0.04) |
| w/ Activation | 18.6 | 22.7 | 0.32 | 0.41 (0.98) | 0.41 (0.99) | 82.57 (1.00) | 0.50 (0.73) |
| Reference Value | 16.5 | 15.6 | 0.31 | 0.77 (1.00) | 0.46 (1.00) | - | - |

(b) Results of the level 2 and level 3 tasks.

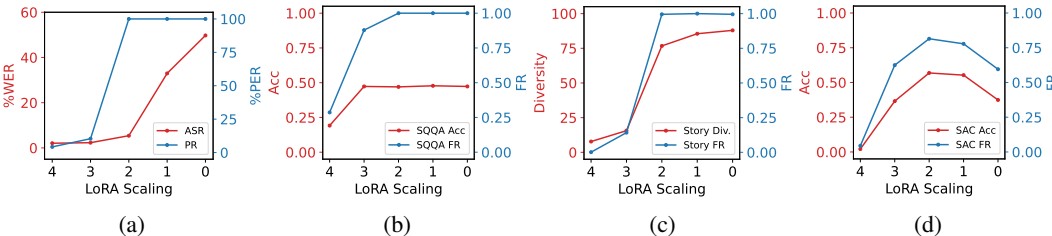

| (a) | (b) | (c) | (d) |

Figure 2: Performance changes on ASR & PR (a), SQQA (b), Story (c) and SAC (d) along with the FR of the emergent abilities against the number of training steps during activation tuning.

with cross-modal reasoning abilities, which, together with the drops in %PER, proves the existence of the intrinsic conditional LM embedded in LoRA.

Figure 3: Performance changes on ASR & PR (a), SQQA (b), Story (c) and SAC (d) together with the FR of the emergent abilities when discounting the LoRA scaling factor at test time.

## 5.3 ANALYSIS OF TASK OVER-FITTING AND ACTIVATION TUNING

To verify the influences of the intrinsic conditional LM, we calculate the perplexities (PPLs) of $P_{\Lambda}(\mathbf{Y}|\mathbf{X}, \mathbf{I})$ and $P_{\Lambda}(\mathbf{Y}|\mathbf{X})$ of each step of the activation tuning stage. In detail, for a given audio $\mathbf{X}$, the probability of the $\mathbf{Y}$ sequence corresponding to the probed task is calculated using teacher forcing based on the text instruction prompt $\mathbf{I}$ of a certain task or without using any $\mathbf{I}$.

As shown in Fig. 4, PPLs were compared between the ground-truth responses of the task of Story/SAC (generated by discounting the LoRA scaling factor) and the responses of the task that the model performed incorrectly due to *task over-fitting* (AAC in this example). In sub-figures (a) and (b), no text instruction prompt was used when computing $P_{\Lambda}(\mathbf{Y}|\mathbf{X})$, showing that without activation tuning the PPL of the $\mathbf{Y}$ of AAC was obviously lower than that of the $\mathbf{Y}$ of Story/SAC. During activation tuning, the PPL gaps between these tasks were mitigated, meaning the bias to the dominant AAC task was considerably reduced. In sub-figures (c) and (d), we probed $P_{\Lambda}(\mathbf{Y}|\mathbf{X}, \mathbf{I})$, the PPLs with the instructions of Story and SAC respectively. It is revealed that before activation tuning, the PPL of AAC is lower than that of Story/SAC even if the text instruction prompt is to perform Story/SAC, which explains the failure of the model to follow the instruction. During activation

tuning, the PPL values of the $\mathbf{Y}$ of Story/SAC gradually become lower than those of AAC, and the model can eventually perform the instructed task.

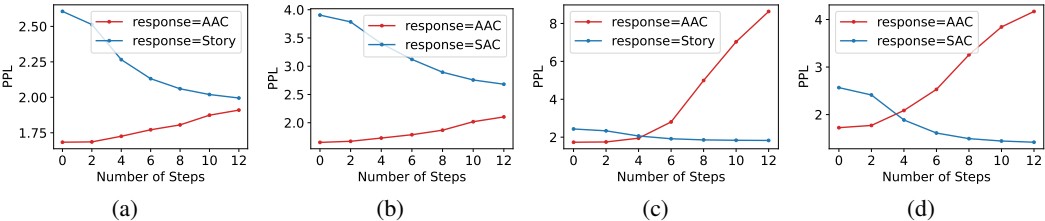

(a)          (b)          (c)          (d)

Figure 4: Changes in PPL during the activation tuning stage. (a) shows $-\ln P_{\mathbf{\Lambda}}(\mathbf{Y}|\mathbf{X})$ for AAC and Story, (b) shows $-\ln P_{\mathbf{\Lambda}}(\mathbf{Y}|\mathbf{X})$ for AAC and SAC, (c) shows $-\ln P_{\mathbf{\Lambda}}(\mathbf{Y}|\mathbf{X},\mathbf{I} = \text{Story})$, and (d) shows $-\ln P_{\mathbf{\Lambda}}(\mathbf{Y}|\mathbf{X},\mathbf{I} = \text{SAC})$.

## 5.4 ACTIVATION TUNING USING DIFFERENT TASKS AND DATA

We have explored several activation methods to activate the model, including training on long stories written based on the audio, text-based question-answering (QA) task pairs with long answers, and the ASR task with long speech transcriptions. Both LoRA and Q-Former are fine-tuned with these three methods. By ignoring the Q-Former and audio inputs, Vicuna and LoRA are fine-tuned as an adapted text-based LLM. As another control group, the experiment of using both LoRA and Q-Former but without tuning the Q-Former is also conducted.

The results are shown in Table 4. From the results, neither ASR (Long), training ASR with long labels, nor Story (Text based), training LoRA based on only text prompt input, can activate the model. Training with stories or QA with long answers, Story or QA (Long), can. This lies in the fact that the ASR task emphasises improving high-quality cross-modal alignments that make the distribution even more biased towards the intrinsic conditional LM. As for text-based fine-tuning on stories, this actually affect $P(\mathbf{Y}|\mathbf{T}_{\mathrm{x}},\mathbf{I})$ instead of $P(\mathbf{Y}|\mathbf{X},\mathbf{I})$, where $\mathbf{T}_{\mathrm{x}}$ is the reference text of the speech. Therefore, text-based training cannot alleviate task over-fitting. QA data with long answers can activate the LLM, but the performance of the activated model is worse than that activated using the Story task, which especially results in a higher repeat rate of the response generation. That is possibly due to the fact that the paired answer to the QA task is less diverse than the Story task, which leads to less effective regularisation of the intrinsic conditional LM $P_{\mathbf{\Lambda}}(\mathbf{Y}|\mathbf{X})$.

Table 4: Results of selected tasks with activation tuning performed based on different tasks. "Repeat Rate" is the percentage of test samples that SALMONN generated responses repeatedly.

| Activation Method | ASR↓ | PR↓ | SQQA↑ | Story↑ | SAC↑ | Repeat Rate↓ |
|---|---|---|---|---|---|---|
| w/o Activation | 2.1 | 4.2 | 0.19 (0.29) | 7.77 (0.00) | 0.02 (0.04) | 0.2% |
| Story | 2.1 | 4.2 | 0.41 (0.98) | 82.57 (1.00) | 0.50 (0.73) | 0.1% |
| QA (Long) | 2.1 | 4.3 | 0.40 (0.93) | 59.82 (1.00) | 0.34 (0.71) | 4.6% |
| ASR (Long) | 2.2 | 4.2 | 0.22 (0.28) | 7.87 (0.00) | 0.12 (0.03) | 0.1% |
| Story (Text based) | 2.1 | 4.2 | 0.23 (0.32) | 8.45 (0.03) | 0.11 (0.03) | 0.1% |
| Story (LoRA only) | 2.1 | 4.2 | 0.44 (0.96) | 82.29 (1.00) | 0.34 (0.65) | 0.2% |

## 6 CONCLUSION

This work proposes SALMONN, a speech audio language music open neural network that can be regarded as a step towards generic hearing abilities for LLMs. Equipped with dual auditory encoders, SALMONN achieved competitive performances on trained tasks including speech recognition, audio captioning and speech translation *etc.*, while generalising to a range of untrained understanding tasks such as slot filling, speech translation for untrained languages and keyword extracting. Moreover, a proposed activation tuning stage enables SALMONN with remarkable emergent abilities, such as audio-based storytelling and speech audio co-reasoning. As a result, with thorough and comprehensive experimental evaluations, SALMONN has demonstrated a promising direction for developing generic hearing AI in the future.

# 7    REPRODUCIBILITY STATEMENT

To make the experiments and models reproducible, the training data and the benchmark details are provided in Section 4.2 and Section 4.3. The source code, model checkpoints, and data are released on the SALMONN project GitHub page.

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

## A  EXAMPLES OF SALMONN

SALMONN is capable of a large range of auditory tasks. Here we list some examples in Fig. 5-21.

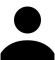 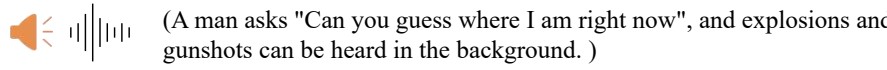 (A man asks "Can you guess where I am right now", and explosions and gunshots can be heard in the background. )

**USER:** Recognize the speech and give me the transcription.

Can you guess where I am right now?

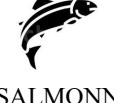

SALMONN

Figure 5: Automatic Speech Recognition

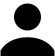 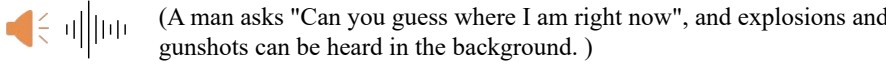 (A man asks "Can you guess where I am right now", and explosions and gunshots can be heard in the background. )

**USER:** Listen to the speech and translate it into German.

Kannst du erraten, wohin ich gerade fahre?

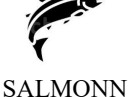

SALMONN

Figure 6: Automatic Speech Translation (En2De)

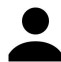 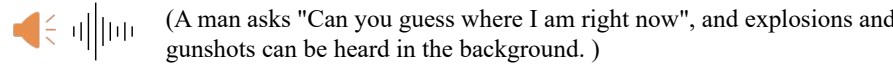 (A man asks "Can you guess where I am right now", and explosions and gunshots can be heard in the background. )

**USER:** Provide the phonetic transcription for the speech.

sil K AH0 N Y UW1 G EH1 S W EH1 R AY1 EY1 EH1 M R AY1 T N AW1 sil

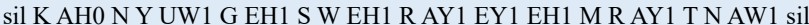 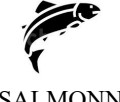

SALMONN

Figure 7: Phone Recognition

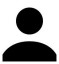 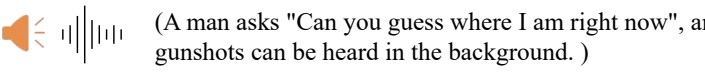 (A man asks "Can you guess where I am right now", and explosions and gunshots can be heard in the background. )

**USER** Please describe the audio.

A person is speaking while gunshots are being fired in the background. 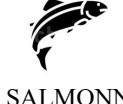

**SALMONN**

Figure 8: Automatic Audio Captioning

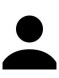 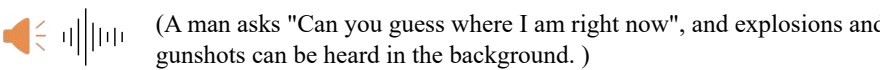 (A man asks "Can you guess where I am right now", and explosions and gunshots can be heard in the background. )

**USER** Recognize what the speaker says and describe the background audio at the same time.

The speaker says "Can you guess where I am right now?" while the background audio has gunshots and explosions. 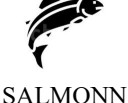

**SALMONN**

Figure 9: Joint Speech and Audio Captioning

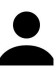 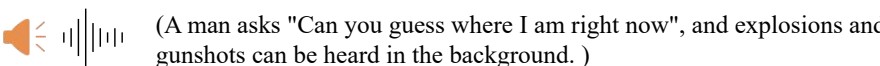 (A man asks "Can you guess where I am right now", and explosions and gunshots can be heard in the background. )

**USER** Please answer the speaker's question in detail based on the background sound.

Based on the background sound, it seems like the speaker is in a war zone or a combat situation. The sound of gunfire and explosions can be heard in the background. The speaker is asking if the listener can guess where they are. 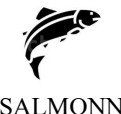

**SALMONN**

Figure 10: Speech Audio Coreasoning

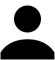 (Birds chirping, a woman singing, a man speaking and a duck quacking.)

**USER:** Please list each event in the audio in order.

1. Birds chirping in the distance.
2. A woman singing.
3. A man speaking.
4. A duck quacking.

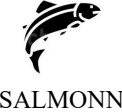

SALMONN

Figure 11: Audio Event Detection

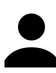 (Birds chirping, a woman singing, a man speaking and a duck quacking.)

**USER:** Based on the audio, write a story in detail. Your story should be highly related to the audio.

It was a beautiful day in the countryside. The sun was shining, the birds were singing, and the wind was blowing gently through the fields. In the midst of all this natural beauty, a group of ducks were swimming in a pond.

As they swam, one of the ducks began to quack in a unique way. It was as if the duck was trying to say something important. The other ducks noticed this and began to listen.

Suddenly, a man appeared on the edge of the pond. He had heard the duck's quacking and had come to see what was going on. The duck continued to quack, and the man began to speak ......

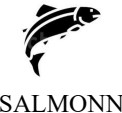

SALMONN

Figure 12: Audio Story Telling

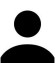 (Birds chirping, a woman singing, a man speaking and a duck quacking.)

**USER:** How many speakers did you hear in this audio? Who are they?

I heard two speakers in this audio. One is a man and the other is a woman.

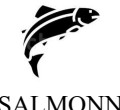

SALMONN

Figure 13: Speaker Recognition

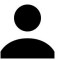 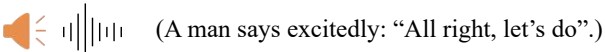 (A man says excitedly: "All right, let's do".)

**USER:** Describe the emotion of the speaker.

The speaker is excited. 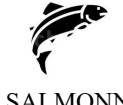

**SALMONN**

Figure 14: Emotion Recognition

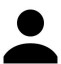 (A person asks "What is the highest mountain in the world".)

**USER:** Please answer the question in detail.

The highest mountain in the world is Mount Everest, located in the Himalayas between Nepal and Tibet. It stands at a height of 8,848 meters (29,029 feet) above sea level. 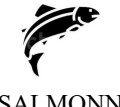

**SALMONN**

Figure 15: Spoken-query-based Question Answering

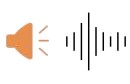 (A part of Steve Jobs' speech: "Your time is limited so don't waste it living someone else's life. Don't be trapped by dogma which is living with the results of other people's thinking. Don't let the noise of other's opinions drown out your own inner voice. And most important have the courage to follow your heart and intuition. They somehow already know what you truly want to become everything else is secondary. When I was young there was an amazing publication called the whole earth catalog, which was one of the bibles of my")

**USER:** Give me only three keywords of the text. Explain your reason.

The keywords are 'time', 'limited', and 'courage'.

The reason for choosing these keywords is that the speaker emphasizes the importance of not wasting time living someone else's life, not being trapped by dogma, and having the courage to follow one's own inner voice. 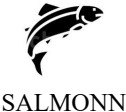

**SALMONN**

Figure 16: Keywords Extracting

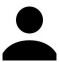 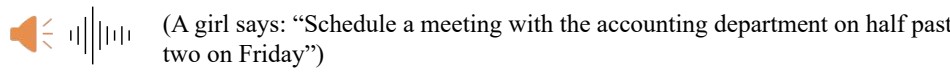 (A girl says: "Schedule a meeting with the accounting department on half past two on Friday")

**USER**: What is the time mentioned in the speech?

Half past two on Friday 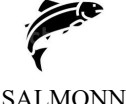

**SALMONN**

Figure 17: Slot Filling

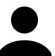 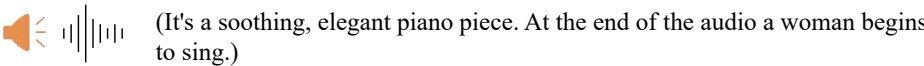 (It's a soothing, elegant piano piece. At the end of the audio a woman begins to sing.)

**USER**: Please describe the music in detail.

This is a beautiful piano and vocal piece. The piano plays a simple melody while the female vocalist sings in a high register. The atmosphere is romantic and dreamy. The piece has a soft and mellow feel to it. This piece could be used in a romantic movie scene or as a lullaby. 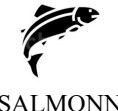

**SALMONN**

Figure 18: Music Captioning

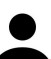 (It's a soothing, elegant piano piece. At the end of the audio a woman begins to sing.)

**USER**: What is the emotion of the music? Explain the reason in detail.

The music is emotional and evokes feelings of love and affection. The reason for this is that the music is a love song, with lyrics that express the deep emotions and feelings of love and affection. The music is also instrumental, which means that it relies solely on the melody and harmony of the instruments to convey these emotions. This creates a more intimate and personal experience for the listener, allowing them to interpret the music in their own way and connect with it on a deeper level. 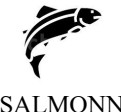

**SALMONN**

Figure 19: Music Emotion Recognition

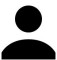 (It's a soothing, elegant piano piece. At the end of the audio a woman begins to sing.)

**USER:** Can you write some lyrics of the song?

Sure, here are some lyrics for the song:

Verse 1:
You're the one I've been waiting for,
My heart beats faster when I'm near you,
You're the one I've been searching for,
My love for you will never fade.

Chorus:
You're perfect, you're flawless, ...

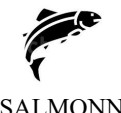

**SALMONN**

Figure 20: Lyrics Writing

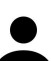 (It's a soothing, elegant piano piece. At the end of the audio a woman begins to sing.)

**USER:** Give me a title of the music based on its rhythm and emotion.

The music is a beautiful piano piece with a slow and emotional rhythm. It is a love song with a female vocalist singing in a soft and gentle tone. The title of the music could be something like 'Forever Love' or 'Endless Emotion'.

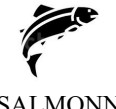

**SALMONN**

Figure 21: Titling Music

## B PERFORMANCE ANALYSIS

As shown in Table 3, with activation tuning[2], SALMONN

1. performs close to the state-of-the-art results on the trained tasks including ASR, En2Zh, AAC and MC;
2. learns to handle PR and OSR tasks which are difficult for the cascaded approach of "Vicuna + Whisper";
3. generalises well on a wide range of speech-grounded NLP tasks, especially such as En2De and En2Ja where SALMONN performs better than the "Whisper+Vicuna" cascaded system, which indicates the superior of the audio-grounding system to cascade system on avoiding error propagation and the loss of non-linguistic information (*e.g.* prosody);
4. tackles tasks such as audio-based storytelling and SAC which existing models can not handle to our best knowledge.

---

[2]Added at the requests of anonymous reviewers. We thank the reviewers for the suggestion.

Table 5: Mean & standard deviation (in brackets) of the output sequence perplexity (PPL).

| Model Checkpoint | ASR | AAC | PR | Story | SAC |
|---|---|---|---|---|---|
| Pre-training | 1.1 ($\pm$0.078) | 5.2 ($\pm$0.42) | 270 ($\pm$60) | 2.9 ($\pm$0.051) | 5.4 ($\pm$0.69) |
| LoRA removed | 5.9 ($\pm$1.5) | 40 ($\pm$2.0) | 100 ($\pm$1.7) | 2.2 ($\pm$0.028) | 2.7 ($\pm$0.18) |
| Instruction Tuning | 1.1 ($\pm$0.054) | 5.0 ($\pm$0.56) | 1.1 ($\pm$0.015) | 2.4 ($\pm$0.030) | 2.6 ($\pm$0.066) |
| LoRA removed | 5.7 ($\pm$1.3) | 40 ($\pm$3.3) | 92 ($\pm$8.2) | 2.2 ($\pm$0.023) | 2.2 ($\pm$0.087) |
| Activation Tuning | 1.1 ($\pm$0.048) | 6.3 ($\pm$0.80) | 1.1 ($\pm$0.015) | 1.8 ($\pm$0.026) | 1.4 ($\pm$0.0046) |

The underlying reason for using the cascaded Whisper + Vicuna system for reference values of the level 2 tasks lies in the fact that all level 2 tasks are zero-shot and there is no other audio-grounding system apart from SALMONN that can perform such tasks as zero-shot. An advantage of SALMONN over the cascaded system is that SALMONN can leverage both linguistic and paralinguistic information required for spoken language understanding.

Despite such advantages, SALMONN has performance limitations on some tasks.

1. First, PR is achieved by extending the LLM to consider phonemes as a new writing system. Since recognising phonemes requires finer-grained modelling of pronunciation information than recognising the word pieces used by the original Whisper ASR, it is not easy for the SALMONN model built upon an existing Whisper speech encoder to perform as well as a specialised model on the PR task.

2. Similarly, SALMONN has a relatively high WER on OSR since the Whisper ASR was not able to perform OSR.

3. The success of SQQA mainly relies on the understanding of the spoken questions (*e.g.* "What is the highest mountain in the world") and answering the questions based on the commonsense knowledge stored in the text-based LLM. The drop in SQQA performance indicates that the use of LoRA cross-modal adaptation may cause the LLM to "forget" some text-based commonsense knowledge.

## C  STATISTICS OF TASK OVER-FITTING

In this section, we analyse the changes of perplexity (PPL) in the three training stages to shed light on the underlying principle of activation tuning. As shown in Table 5, The PPL of ASR and AAC tasks come to very small values after the first pre-training stage, revealing that the model has learned cross-modal alignment. The PPL of PR drops after the instruction tuning stage since PR relies on LoRA to learn the output tokens of phonemes as a new "language". While the PPLs of Story and SAC also drop after instruction tuning, they are still not small enough for the tasks to be successfully performed, unless LoRA is removed at test time or an additional activation tuning stage is performed. Moreover, unlike the removal of LoRA, the PPLs on the ASR, AAC, and PR tasks remain almost unchanged after activation tuning, showcasing the advantages of this approach.

## D  CHANGES IN PERPLEXITY DURING TRAINING

Changes in perplexity during the proposed three-stage training on six tasks including ASR, AAC, AST (En2Zh), OSR, Story, and SAC are visualized in Fig. 22.

## E  CALCULATION OF FOLLOWING RATE (FR) OR ACCURACY (ACC) FOR SQQA, SF, STORY AND SAC TASKS

For the SQQA and SF tasks, if the WER calculated between model output and the question in the speech is less than 30%, it is regarded that the model goes for ASR and does not follow instructions. For the Story task, we set the max length of output tokens to 200. Under this setting, answers shorter than 50 words are regarded as disobeying the instructions, and we count the number of different

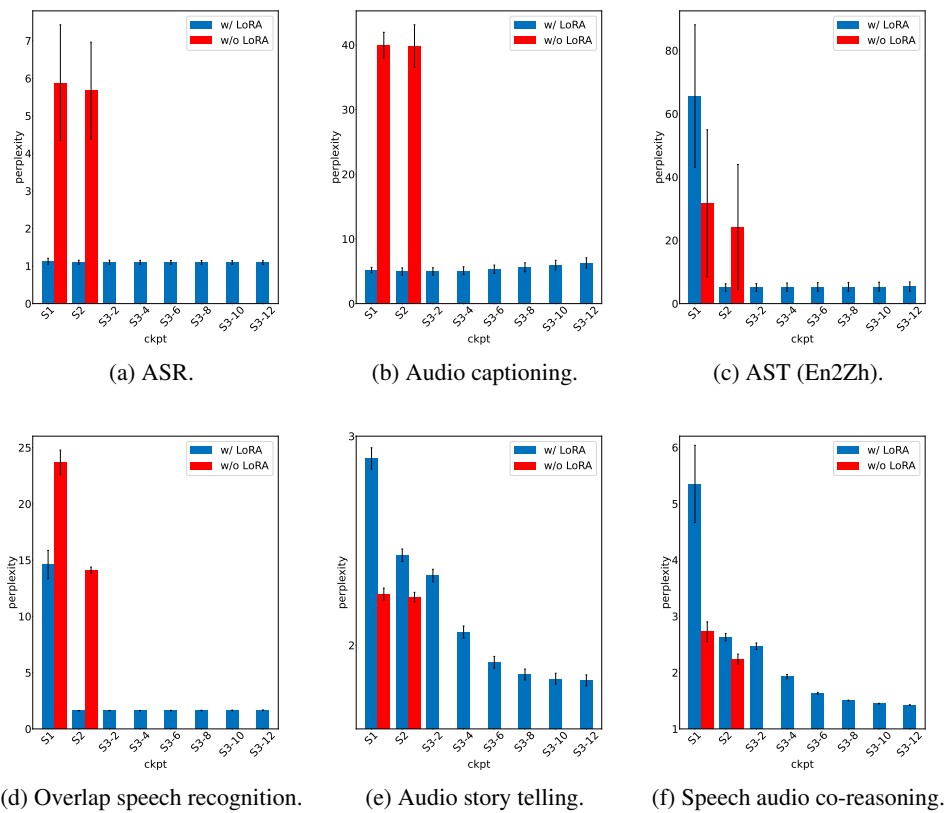

Figure 22: Changes in perplexity during the three-stage training. S1 is cross-modal pre-training stage, S2 is instruction tuning stage and S3-$t$ is the $t^{th}$ step in the activation tuning stage.

words in the story to represent the diversity metric for storytelling. For the SAC task, we use GPT-3.5 to determine whether our model follows the instruction or answers the question correctly based on the background audio caption and the question.

## F    PROMPT TEMPLATE FOR SALMONN

To train SALMONN to generate responses to a text instruction given an audio input, we utilize a prompt template as follows:

*USER:* **[Auditory Tokens]** *Text Prompt \n ASSISTANT:*

The "**[Auditory Tokens]**" are the output variable length text-like tokens of the window-level Q-Former. Noted that this template matches the one used for training Vicuna (Chiang et al., 2023).

## G    PROMPTS INTERACTING WITH GPT3.5

In this work, we utilize GPT3.5 to help us generate some data and evaluate the quality of the model output automatically. We list our prompts for different purposes in Table 6. Words with all capital letters in the prompt will be replaced with the appropriate description before being fed into GPT3.5.

## H  POTENTIAL TO EXTEND SALMONN TO SPEECH AND AUDIO GENERATION

Although SALMONN is designed to focus on enabling LLMs with hearing abilities, it is possible to extend SALMONN to speech generation.[3] The human speech production mechanism is related to auditory perception. A well-known phenomenon attributed to the speech chain is the "Lombard reflex" which describes the effect where individuals raise their voice level to be heard more clearly while speaking in noisy environments (Lane & Tranel, 1971). This reveals the importance of having generic hearing abilities when empowering AI with fully human-like speaking abilities, and the opportunities in text-to-speech (TTS) synthesis enabled by SALMONN. This also matches the recent development in TTS that the text and audio contexts from the surrounding utterances are useful to achieve more natural prosody modelling and enable the use of more natural and casual speech data (Xu et al., 2021; Guo et al., 2021; Oplustil-Gallegos et al., 2021; Zhang et al., 2023c).

---

[3]Added at the request of an anonymous reviewer. We thank the reviewer for the suggestion.

| Purposes | Prompts |
|---|---|
| To generate audio QA data given audio caption text. | Below I will give you some sentences that you will need to help me generate **only one** question, and its corresponding answer. These sentences are captions of some audio. Your question should be highly related to the audio caption, and your answer must be **correct**, and should be simple and clear. \n Your response should strictly follow the format below: \n {"Question": "xxx", "Answer": "xxx"} \n Here are the sentences: |
| To generate speech QA data given speech recognition text. | Below I will give you some sentences that you will need to help me generate **only one** question, and its corresponding answer. Your question should be highly related to the sentences, and your answer must be **correct**, and should be simple and clear. \n Your response should strictly follow the format below: \n {"Question": "xxx", "Answer": "xxx"} \n Here are the sentences: |
| To evaluate answers of the model of spoken-query-based question answering (SQQA). | Next I will give you a question and give you the corresponding standard answer and the answer I said. You need to judge whether my answer is correct or not based on the standard answer to the question. I will give you the question and the corresponding answer in the following form: {'Question': 'xxx', 'Standard Answer': 'xxx', 'My Answer': 'xxx'} \n You need to judge the correctness of my answer, as well as state a short justification. Your responses need to follow the Python dictionary format: \n {"Correct": True / False, "Reason": "xxx"} \n Now, I will give you the following question and answer: SENTENCEHERE \n Your response is: |
| To evaluate whether the model attempts to do the speech audio co-reasoning (SAC) task. | There is an audio clip, and there is a person in the audio asking questions. I now have an AI model that needs to go and answer the speaker's question based on the background audio. I'll tell you the question the speaker is asking the output of my AI model, and what you need to determine: whether my AI model is trying to answer the question and why. You need to be especially careful that my model may just be describing the audio without hearing your question and answering it. You don't need to care about the correctness of the answer. All you need to focus on is whether the model is trying to answer the question. Your response needs to follow the format of the python dictionary: {"Response": "Yes/No", "Reason": "xxx"}.\n Question in audio: <QUESTION> \n Model Output: <OUTPUT> \n Your Response: |
| To evaluate whether the model successfully completes the SAC task. | There is an audio clip, and there is a person in the audio asking questions. I now have an AI model that needs to go and answer the speaker's question based on the background audio. I'll tell you the question asked by the speaker, some description of the background audio, and the output of my AI model, and you need to decide whether my AI model answered it correctly, and why. Your response needs to follow the format of the python dictionary: {"Response": "Yes/No", "Reason": "xxx"}.\n Question in audio: <QUESTION> \n Background Audio: <AUDIO> \n Model Output: <OUTPUT> \n Your Response: |

Table 6: Purposes and prompts of using GPT3.5.

