# OpenReview forum: "SALMONN: Towards Generic Hearing Abilities for Large Language Models"
_ICLR.cc/2024/Conference — ICLR 2024 poster_

### Official Review · Reviewer_bK7i · 2023-10-28

**Soundness:** 2 fair
**Presentation:** 2 fair
**Contribution:** 3 good
**Rating:** 6
**Confidence:** 4

**Summary:**

The authors propose to fuse both speech encoder (whisper) and general audio encoder (beats) as inputs, connect with LLM via a Q-Former and fine-tune with LoRA. Besides pre-training and instruction tuning stages, the authors also propose an activation tuning stage, which is to prevent from overfitting to short captions and is able to generate long and diverse stories. For some of the instruction tuning dataset, the authors leverage LLM for curation.

**Strengths:**

- The combination of speech and audio encoders are interesting ideas and the results of speech audio co-reasoning provide new capabilities for audio understanding.

**Weaknesses:**

- According to the results in table 3, it seems that the proposed method only works significantly better on level 3 tasks of Story and SAC, which the evaluation metrics are specifically designed and there is no other reference value from other models provided. It would be better to provide more information on the 2 tasks and providing other baseline performance on these two tasks.
- For the Story task, it is worth including accuracy FR along with diversity FR. This can provide a more holistic understanding of the tradeoffs.
- For some tasks in level 1 and 2, the performance of proposed method is significant worse, e.g. PR, OSR. and SQQA for level 2. It might worth providing some in depth discussion and analysis. For example, can it be that adding both whisper and beats features introduce more confusion to the model? How are the Q-former attending to the concatenated features to make predictions?

**Questions:**

- The process of activation tuning stage is not clear described, according to the last paragraph of 4.2, if the data is generated by SALMONN model, what do they look like? It would be helpful to provide an example. Also for teacher-forcing training, is it just a standard cross-entropy loss on generated text? How do you control the diversity and length of generated examples?
- How are the instruction prompted for evaluation tasks? How do you instruct the model for evaluating these tasks?
- Is ChatGPT also leveraged to generate text data for some evaluation tasks, especially for SAC and Story. If so, how are they leveraged to curate answers?
- How are the prompted QA for training generated by ChatGPT verified?

---

> ### Author Response · Authors · 2023-11-15
>
> We thank the reviewer for the positive comments on the structure of SALMONN. We have revised parts of the paper (marked in yellow) based on comments from all reviewers. Regarding your constructive suggestions, we will respond to them one-by-one as follows.
> - Weakness 1:
>   - To the best of our knowledge, SALMONN is the first end-to-end model that is capable of audio-based storytelling and speech audio co-reasoning (SAC). We did not find a model which would even attempt to do both tasks. Therefore, we can not provide reference values for these tasks.
> - Weakness 2:
>   - Sorry that we are confused by this question, since it is not clear to us what "accuracy FR" and "diversity FR" mean? In this paper, the following rate (FR) is calculated as the chance that the model follows the user instructions, which is a different metric compared to accuracy and diversity.
>   - Regarding storytelling, since it is a zero-shot text generation task, which is highly subjective, we are not able to test it's accuracy in the traditional way. However, we evaluate a new metric "events coverage" to fulfill the reviewer's request, which is calculated as $\frac{\text{num audio events appear in the story}}{\text{num audio events appear in the audio}}$. SALMONN achieves an event coverage rate of 0.5642.
> - Weakness 3:
>   - This need to be explained to some extent. We achieve PR by extending the LLM to use phonemes as a new writing system. Since recognising phonemes require finer-grained modelling of pronunciation information than recognising the word pieces used by the original Whisper ASR, it is reasonable that the SALMONN model built upon an existing Whisper speech encoder cannot perform very well on PR, compared to a specialised PR model. This is also the reason for the relatively low performance of OSR, since the Whisper ASR was not able to achieve overlapping speech recognition.
>   - The success of SQQA mainly relies on the understanding of the spoken questions (e.g. "What is the highest mountain in the world") and answering the questions based on the commonsense knowledge stored in the text-based LLM. The drop on SQQA performance indicates that the use of LoRA cross-modal adaptation may cause the LLM to "forget" some text-based commonsense knowledge.
>   -  In terms of the Q-Former usage, we use the Q-Former with a sliding window, which converts the features in the ~0.33 second window into one text-like token. The length of such a window is close to the time it takes to say a word.
> - Question 1:
>   - An example of the generated story is shown at Figure 12 in Appendix A in the paper.
>   - You are also welcome to experience it with our demos at the link: https://github.com/the-anonymous-account/SALMONN, using the text prompt: Based on the audio, write a story in detail. Your story should be highly related to the audio.
>   - For teacher-forcing training, it is the standard cross-entropy loss on the generated text. This is the same loss we've been using throughout training.
>   - Actually, we did not purposely control the diversity and the length of the model's output. We just activate the model so that it can utilize the power of the LLM and generate very diverse answers.
> - Question 2:
>   - Here are some examples of prompts for evaluating different tasks
> | Level | Task | Text Prompt |
> | --- | --- | --- |
> | I  | ASR | Recognize the speech and give me the transcription. |
> |  | AAC | Please describe the audio. |
> |  | ER | Describe the emotion of the speaker in one word. |
> | II | KE | Give me only three keywords of the text. |
> |  | SQQA | Please answer the question in detail. |
> | III | Story | Based on the audio, write a story in detail. Your story should be highly related to the audio. |
> |  | SAC | Use your strong reasoning skills to answer the speaker's question in detail based on the background sound. |
>
>   - Whether it's training or testing, we prompt the model using the same format: `USER: [Auditory Tokens] Text Prompt \nASSISTANT:`. The [Auditory Tokens] are the output of the QFormer.
> - Question 3:
>   - For SAC and Story, we did not use ChatGPT to generate text labels. In fact, we did not use any labels for the two tasks, as there is no single correct answer for them.
>   - For SAC, we use ChatGPT to check whether SALMONN's output is reasonable given the question, the background audio caption, and SALMONN's output. We also perform human checks to validate the correctness of ChatGPT evaluations. Regarding Storytelling, we only calculate the diversity in the paper and did not use ChatGPT for evaluation.
> - Question 4:
>   - The QA samples are generated based on the references of ASR transcription / audio caption / music caption using ChatGPT. We manually checked some of the generated samples and they are all reasonable.
>   - In addition, using ChatGPT to generate training data is now a common approach. For example, WavCaps, which we use for pretraining audio captioning, is a ChatGPT-Assisted Weakly-Labelled dataset.

---

> > ### Comment · Reviewer_bK7i · 2023-11-18
> >
> > Thank the authors for your responses. I would encourage the authors to integrate some of the explanations (especially weakness 3) into the final version. This helps the reader to understand the tradeoffs. I am increasing my ratings.

---

> > > ### Author Response · Authors · 2023-11-19
> > >
> > > Thank you very much for increasing the score and your constructive suggestions! We have added analysis of the model's performance on each task in the revised paper. Due to page limit, we put it at appendix B and point it in section 5.1.

---

### Official Review · Reviewer_RbXH · 2023-10-29

**Soundness:** 3 good
**Presentation:** 2 fair
**Contribution:** 3 good
**Rating:** 8
**Confidence:** 3

**Summary:**

This paper presents SALMONN, a novel method for equipping LLMs with hearing abilities by leveraging additional adapter modules and LoRA weights to train the LLM on a range of speech, audio, and music understanding tasks.

**Strengths:**

- The overall architecture design is quite clever, and its use of generally open source models and datasets is very important to the audio-text field as a whole.
- The amount of time spent towards analyzing and mitigating the models failure modes is quite useful, and the authors provide an incredibly detail analysis of model behavior under different finetuning configurations.

Despite some of the concerns mentioned below (which mostly address overall clarity rather than content), the paper presents an incredibly detailed analysis of a novel architecture and how to adapt LLMs for audio-based reasoning, and thus I recommend acceptance.

**Weaknesses:**

- The explanation of the pretraining stage could use a bit more depth. Namely, how is it that the Q-Former and LoRA weights are actually trained during the pretraining stage?
- I think in general, the sections on task-overfitting and activation tuning are relatively hard to parse reading-wise and could be simplified. Unless I am misunderstanding something, task-overfitting is simply the idea that SALMONN overfits to the overrepresented tasks in the dataset, which I think the math in section 3.2 overcomplicates.
- It is hard to tell in the ablations (5.2-5.4) what is being held fixed and what training configurations are being used. Namely, in section 5.2 are the results on the reduced LoRA scaling factor done with or without activation tuning?
- The authors claim that the Level 3 tasks are harder directly, but given the myriad of evaluation metrics for each task it's hard to tell *why* these tasks are necessarily in their own class. Is there some way to show how these tasks are by nature an entirely more difficulty class of problems? Especially as SAC seems to be evaluated by ChatGPT outputs, it is hard to tell much about the actual performance of the model.

**Questions:**

- What is "monotonic alignment" as mentioned in the Q-Former section?

---

> ### Author Response · Authors · 2023-11-15
>
> We would first like to thank the reviewer for the positive and constructive feedback and the recommendation for acceptance. We also revised parts of the paper (marked in yellow) based on comments from all reviewers. And we will respond to your constructive suggestions one-by-one below.
> - Weakness 1:
>   - The training process of the pretraining stage is almost the same as the instruction tuning stage, except that the data is different. We will make this clear in the updated version of the paper. In pretraining, both the Q-Former and LoRA are trained, and we use text prompts to control the model to do either ASR or AAC. These two tasks are used in pretraining since they provide direct alignments between the sounds of speech contents and audio events with their text descriptions.
> - Weakness 2:
>   - Thanks for your advice. Task over-fitting is indeed the fact that the model overfits the dominant tasks during training. We revised this section to be more concise and intuitive. We also revised Section 5.3 with more intuitive analysis and a new Figure 4, and moved the previous Section 5.3 and Table 4 to Appendix B.
> Here we give a brief explanation about task over-fitting. Using the following equation $P_{\mathbf{\Lambda}}(\mathbf{Y}|\mathbf{X},\mathbf{I})=\frac{P_{\mathbf{\Lambda}}(\mathbf{Y}|\mathbf{X})P_{\mathbf{\Lambda}}(\mathbf{I}|\mathbf{X},\mathbf{Y})}{P_{\mathbf{\Lambda}}(\mathbf{I}|\mathbf{X})}$ (same as Equ. 3 in our paper), we attribute task-overfit to the term $P_{\mathbf{\Lambda}}(\mathbf{Y}|\mathbf{X})$. After training on a large amount of ASR and AAC data (which is necessary for high-quality alignment between audio and text modalities), the intrinsic LM $P_{\mathbf{\Lambda}}(\mathbf{Y}|\mathbf{X})$ has strong bias to these two tasks, which makes the model sometimes violate the instruction prompts and generate responses as if it received an ASR/AAC instruction. After using the proposed activation tuning, $P_{\mathbf{\Lambda}}(\mathbf{Y}|\mathbf{X})$ can achieve a more balanced distribution between the texts that are either strongly or weakly aligned with the audio, and the model can therefore follow the instructions at a higher rate.
> - Weakness 3:
>   - Thank you for pointing this out and we will improve the clarity. When discounting LoRA scaling factor in Section 5.2, the model does not need activation tuning to perform zero-shot cross-modal tasks. This is the most direct way to alleviate task over-fitting, with the cost of considerable performance drops on the trained tasks (e.g. ASR and AAC).
>   - In Sections 5.3 and 5.4, we kept the original LoRA scaling factor and used the activation tuning method to activate the model, which enables the model to perform the zero-shot cross-modal tasks while keeping good performance on the trained tasks.
> - Weakness 4:
>   - The level 2 and level 3 tasks are distinguished primarily based on whether they can be achieved using a  cascade ASR + LLM pipeline. The level 2 tasks are speech-based natural language processing tasks that can be accomplished by first transcribing speech into texts using an external ASR and then feeding the derived texts into a text-based LLM. The level 3 tasks, however, can't be accomplished in this way, since they require the model to fully understand not only speech but also the general sound elements in background audio and music, which are more difficult.
> - Question 1:
>   - Monotonic alignment refers to the alignment of audio signals with their corresponding transcriptions (or caption) in a way that maintains a consistent and non-decreasing relationship between the time frames and the corresponding elements in the audio.
>   - As we use the Q-Former with a sliding window, which converts the features in the ~0.33-second window into one text-like token, the Q-Former output tokens and frames with 0.33-second window are one-to-one. Therefore, we say that the Q-Former output is enforced to have a monotonic alignment with acoustic features.

---

> > ### Comment · Reviewer_RbXH · 2023-11-17
> >
> > The reviewer thanks the author for their response, and based on changes to the draft, has increased the score.

---

> > > ### Author Response · Authors · 2023-11-18
> > >
> > > Thank you for increasing the score! We sincerely appreciate your constructive suggestions and your recognition of this work.

---

### Official Review · Reviewer_KyXG · 2023-11-04

**Soundness:** 3 good
**Presentation:** 3 good
**Contribution:** 2 fair
**Rating:** 6
**Confidence:** 4

**Summary:**

This paper proposed SALMONN, which is a single unified multimodal model to integrate speech and audio encoders with a pre-trained text LLM. The paper shows that SALMONN can achieve competitive performance on a variety of speech tasks used in training, including ASR, ST, emotion recognition, audio QA, speaker verification, audio and music captioning etc. The paper also studies the capabilities of SALMONN on zero-shot capabilities such as ST on untrained languages, SLU, SQA, audio-based story telling, speech-audio co-reasoning etc. The paper also explores a new few-shot activation tuning approach.

**Strengths:**

(1)	Value to the community: The proposed SALMONN model unifies modeling a wide variety of speech, audio, and music perception and understanding tasks into a single framework, which is a useful step towards the research for AGI.

(2)	The innovation of the paper is mostly in choosing and piecing together existing approaches for this unified framework, using and evaluating a diverse set of speech/audio/music tasks for pre-training/instruction-finetuning, studying emergent capabilities, analyzing task-overfitting issue and exploring cheap activation tuning to alleviate catastrophic forgetting to training tasks.  SALMONN reasonably adopts several existing approaches. For speech and non-speech audio encoding, SALMONN integrates a speech encoder from Whisper and a BEATS audio encoder. SALMONN uses a Q-Former to convert encoder output to audio tokens for the text LLM (Vicuna in this work). LoRA is applied to align the augmented input space with output space to improve cross-modal alignment for the text LLM.  Following other works, SALMONN used a diverse set of speech, audio, and music tasks in pre-training and instruction finetuning of Q-Former and LoRA.  Notably,  this paper analyzed the task overfitting issue and provided insights for activation tuning to alleviate catastrophic forgetting from instruction tuning. The paper also studies the capabilities of SALMONN on handling cross-modal emergent tasks.

(3)	Overall, the paper is clearly written.

(4)	Empirical evaluations are comprehensive. The three levels are helpful organizations, as  speech tasks used in instruction tuning, unseen speech-based NLP tasks which can effectively evaluate speech-text cross-modal alignments, and the proposed new audio-based story telling and speech audio co-reasoning tasks which require understanding mixture of speech and non-speech auditory information.

**Weaknesses:**

(1)	In empirical validations, the choice of reference values (as shown in Table 2) needs to be clarified, and more importantly, these choices need to be justified. It is not clear which model size is used for Whisper when it is used as reference values. Also, the choice of simply cascading Whisper + Vicuna needs to be justified as the reference value for many tasks, since it may not be as competitive as other E2E models (e.g., recent speech LLMs), including SOTA. Without clear knowledge how strong these reference values are, it is not easy to judge how strong SALMONN performs as shown in Table 3.

(2)	Some key implementation details are missing.  The training data as shown in Table 1 are highly unevenly distributed. It is not clear methods such as data upsampling are used, or batches are designed for multi-task instruction fine-tuning.

(3)	The paper focuses on general hearing capabilities of speech/audio/music. It would be useful to discuss how to extend the model to speech/audio/music generation tasks.

**Questions:**

Please check the comments and concerns raised under Weaknesses.

---

> ### Author Response · Authors · 2023-11-15
>
> We would like to thank the reviewer for highlighting the value of SALMONN to the community that it unifies the wide range of speech/audio/music tasks and is a useful step towards the research for AGI. We would also like to thank the reviewer for the constructive suggestions which will be responded to one-by-one below. Besides, we also revised parts of the paper (marked in yellow) based on comments from all reviewers.
> - Weakness 1: Regarding our choices of reference values, it is only to provide some reasonable references for the readers to understand the tasks and verify the feasibility of our approach.
>   - For all the reference values using Whisper, we use the Whisper Large v2 (1550M), whose encoder is also used to construct the SALMONN model.
>   - Reference values of the AAC, PR, ER, MC and OSR of the Level 1 tasks are state-of-the-art (SOTA) values. Since the encoder of Whisper Large v2 is used in SALMONN, we use its ASR results as the reference value of the ASR task.
> Indeed, the En2Zh translation result evaluated on the CoVoST2-En2Zh test set might not be the most suitable reference value, since SALMONN was trained on the CoVoST2-En2Zh training set while "Whisper + Vicuna" was not. We thank the reviewer for pointing this out and will update the paper using the BLEU scores provided in [1] where the highest BLEU score of En2Zh is 38.9.
>   - Since the Level 2 tasks are mainly speech-grounded natural language processing (NLP) tasks, it is reasonable to use the "Whisper + Vicuna" cascade system to produce the reference values. Since SALMONN treats all Level 2 tasks as zero-shot tasks, the performances of the Vicuna LLM given text input on the NLP tasks are actually the upper bound performance of SALMONN on the Level 2 tasks.
>   - In summary, SALMONN 1) performs close to the SOTA results on the trained tasks including ASR, En2Zh, AAC and MC; 2) learns to handle PR and OSR tasks which are hard for "Vicuna + Whisper"; 3) generalises well on a large range of speech-grounded NLP tasks and 4) tackles tasks such as audio-based storytelling and SAC which existing models can not handle to our best knowledge.
> - Weakness 2:
>   - As shown in Table 1, QA tasks (SQA, AQA, MQA) take the largest portion of the training data (27%), which is to alleviate the task over-fitting problem. Since diverse questions are often used as text prompts for QA tasks, the model becomes less overfitted to common instructions by training on more QA data. We will release our source and training data to provide all implementation details if the paper is accepted.
>   - Since there is an inherent difference in the amount of data available for each task, the portions of training data used for some tasks (e.g. ER and MC) are relatively small. However, despite a very limited amount of training data used for ER and MC, SALMONN can achieve good results on these tasks if the audio and text modalities have been well aligned through the pretraining stage.
>   - As for the design of minibatch, we randomly sample data from the whole training set which means a minibatch is made up of samples of different tasks. We believe this strategy can help produce more robust gradients for updating the parameters, and all the training tasks can be well optimised with enough training steps.
> - Weakness 3:
>   - We thank the reviewer for the advice! We include the discussions about extending SALMONN to speech/audio/music generation in Appendix H in the revised paper. In our humble opinion, SALMONN is very suitable for speech generation due to its generic hearing abilities. There are evidences that human-like natural speech generation builds upon auditory perception. For instance, the "Lombard Effect" (the involuntary tendency of speakers to increase their vocal effort when speaking in loud noise to enhance the audibility of their voice) is a great example showing the connections between hearing and speaking, which is necessary for AGI to achieve completely human-like speech generation.
>
> [1] Wang, C., Wu, A., Gu, J., Pino, J. (2021) CoVoST 2 and Massively Multilingual Speech Translation. Proc. Interspeech 2021, 2247-2251, doi: 10.21437/Interspeech.2021-2027

---

> ### Comment · Reviewer_KyXG · 2023-11-22
> **Still have serious concerns**
>
> Thanks the authors for the responses. However,  I still have serious concerns on the responses to weakness #1: insufficient comparisons of performance between SALMONN and competitive baselines could cause misunderstandings of the performance of the proposed model. For speech emotion recognition, models such as vesper-12 (Chen et al.,2023) should have been compared. For AAC, the ensemble model ( Koizumi et al, 2020) should have been compared. These are just a few examples for various tasks in the paper. Simply comparing to the cascaded Whisper + Vicuna is insufficient, since the pipelined baseline suffers from error accumulation and loss of prosody in ASR output for speech tasks, and may not be as high performing  as other models.   I will keep my score. Thanks.

---

> ### Author Response · Authors · 2023-11-23
> **Response to the concerns of Reviewer KyXG and clarify the key misunderstandings**
>
> We would like to thank Reviewer KyXG for returning to provide further comments. We would like to provide further responses to clarify the misunderstandings and resolve the reviewer's concerns. We are happy to provide comparisons between SALMONN and any existing approach as long as the instruction is clear and leaves the authors sufficient time. In short, the referenced values of ER and AAC in our submission are from audio-grounded end-to-end systems rather than cascaded systems, and the results we quoted are better than the results from the papers suggested by the reviewer.
>
> 1. "For speech emotion recognition, models such as vesper-12 (Chen et al.,2023) should have been compared."
> Regarding speech emotion recognition (ER), the work we compared in our submission is:
>
> W. Wu, C. Zhang, and P.C. Woodland. Emotion recognition by fusing time synchronous and time asynchronous representations. In Proc. ICASSP, 2021.
> We would like to emphasise that this work uses an end-to-end rather than cascaded approach, and achieves considerably better results on IEMOCAP (0.7757 and 0.7841 for 5-fold cross-validation of WA and UA) compared to Vesper-12 (0.707 and 0.708 for 5-fold cross-validation of WA and UA) according to the Table 3 of the Vesper-12 paper:
> W. Chen et al., "Vesper: A Compact and Effective Pretrained Model for Speech Emotion Recognition", 2023.
>
> 2. "For AAC, the ensemble model ( Koizumi et al, 2020) should have been compared. These are just a few examples for various tasks in the paper. "
>
> Regarding automatic audio captioning (AAC), the work we compared in our submission is:
>
> Xinhao Mei et al., "WavCaps: A ChatGPT-assisted weakly-labelled audio captioning dataset for audio-language multimodal research", 2023.
>
> which to our knowledge present more recent and state-of-the-art (SOTA) results on the AudioCaps and Clotho test sets. The results we quoted are produced by an end-to-end system HTSAT-BART. The METEOR | SPIDEr results on AudioCaps are 25.0 | 48.5, and 18.4 | 29.7 on Clotho.
>
> Regarding the (Koizumi et al, 2020) paper the reviewer suggested, we found this paper:
>
> Koizumi et al., "A Transformer-based Audio Captioning Model with Keyword Estimation", 2020.
>
> (Koizumi et al, 2020) provided test set results of 14.9 | 17.7 in terms of METER | SPIDEr on Clotho which are considerably worse than the HTSAT-BART results we quoted from the WavCaps paper (Mei et al., 2023). Therefore, the Referenced Value we currently use for AAC is better than the one the reviewer suggested.
>
> Furthermore, we used the AudioCaps test set for AAC in our submission which is a different test set from the Clotho test set used in (Koizumi et al, 2020).
>
> 3. "Simply comparing to the cascaded Whisper + Vicuna is insufficient, since the pipelined baseline suffers from error accumulation and loss of prosody in ASR output for speech tasks, and may not be as high performing as other models. "
>
> We sincerely agree with the reviewer about this claim, and this is also the motivation for us to develop SALMONN as an audio-grounded end-to-end model.
>
> There are two reasons for using the cascade system for reference values. First, the five tasks En2De (English to German translation), En2Ja (English to Japanese translation), KE (speech keyword extraction), SQQA (spoken-query-based question answering), and SF (slot filling) are evaluated as zero-shot tasks. There is no other audio-grounding system publically available to perform such tasks as zero-shot to our knowledge.
>
> Second, in practice, not only the methodology but also the amount of data used in model construction matters. Both the Whisper ASR and the Vicuna LLM were trained using enormously large amounts of audio and text data, which are more than the audio-text paired data used for any existing audio-grounded end-to-end speech understanding system to our knowledge. Therefore, although we used the cascaded system of Whisper + Vicuna for the five tasks, the referenced values we used are reasonable. However, we are happy to include extra results from audio-grounded systems, if the reviewer could kindly provide some suggestions. To acknowledge the valuable suggestion from the reviewer, we extended Appendix B with additional discussions.

---

### Comment · Area_Chair_tWgC · 2023-11-10
**reviewer-author discussions**

Dear All,

The reviewer-author discussion period will be from Nov. 10 to Nov. 22. For reviewers, please read the authors' responses and acknowledge it, respond to them early on in the discussion, and discuss points of disagreement. Thank you!

AC

---

### Meta-Review · Area_Chair_tWgC · 2023-12-05

**Metareview:**

This paper proposed to enable LLM with the hearing abiliites with the model SALMONN. This is done by using Whipser encoder (for semantic extraction) and BEATs encoder (for acoustic extraction) of the audio signal. Then, the extracted information goes through Window-level Q-Former and then as the input to LLM, with LORA finetuning. The authors designed a three-stage training method that can enable the LLM with the hearing capabilities. It is noted that the authors especially address the task over-fitting challenge, and the activation tuning stage is novel. One strength of this paper is that the authors have carefully designed the tasks for three levels, and showed the effectivess of SALMONN. This is very helpful for future works to follow the same protocol for audio+LLM evaluation.

The main weaknesses:

The major concern is the paper noveltyh. As one reviewer mentioned, the innovation of the paper is mostly in choosing and piecing together existing approaches for this unified framework. Therefore, this paper is more like to show the audience how to make audio+LLM work in an engineering way instead of an algoirthmic way.

Another reviwer had the concern that the proposed method works well on level 3 tasks of Story and SAC, in which the evaluation metrics are specifically designed and there is no other reference value from other models provided. The authors mentioned as the tasks are new, there is no reference point.

Last, the three-stage training seems to be complicated. It is uncertain whether the readers can easily reproduce the strategy in their own tasks.

**Justification For Why Not Higher Score:**

The novelty is the major concern. This paper is more like to show the audience how to make audio+LLM work in an engineering way instead of an algoirthmic way.

**Justification For Why Not Lower Score:**

This investigation is really meaningful to the speech community on how to effectively integrate LLM with audio so that we can perform multiple audio tasks. The designed tasks are also beneficial to the future studies as the reference. The results are also good.

---

### Decision · Program_Chairs · 2024-01-16

Accept (poster)